Intention to use short messaging services for promoting drug adherence among individuals with diabetes in Addis Ababa, Ethiopia

http://orcid.org/0009-0009-4189-6943 Alem Solomon 1 alemsol101@gmail.com
Gulema Hanna 2
1 Epidemiology and Biostatistics, Addis Continental Institute of Public Health (ACIPH) , Addis Ababa , Ethiopia
2 Global Health and Health Policy, Addis Continental Institute of Public Health (ACIPH) , Addis Ababa , Ethiopia
Lounis Mohamed
Electronic publication date: 2024 Nov 13
Publication date: 2024
Volume: 12
Electronic Location ID: e18297
Received 2024 Apr 28; Accepted 2024 Sep 22
Copyright: © 2024 Alem and Gulema
Copyright year: 2024
Copyright holder: Alem and Gulema
License: This is an open access article distributed under the terms of the Creative Commons Attribution License, which permits unrestricted use, distribution, reproduction and adaptation in any medium and for any purpose provided that it is properly attributed. For attribution, the original author(s), title, publication source (PeerJ) and either DOI or URL of the article must be cited.
License URL: https://creativecommons.org/licenses/by/4.0/

Keywords: Diabetes, Adherence, Technology acceptance, Mobile health, Medication compliance, SMS

Funding: The authors received no funding for this work.

==============================
Background

Suboptimal medication adherence among individuals with diabetes presents a significant challenge in low-income nations. Growing evidence demonstrates the effectiveness of text messaging interventions to enhance medication adherence. This study assesses the intention to use Short Messaging Service (SMS) based reminder services in promoting drug adherence among diabetic patients and associated factors in Addis Ababa, Ethiopia.

Methods

An institution-based cross-sectional study was conducted from February 06, 2023, to March 27, 2023, in Addis Ababa, Ethiopia. A sample of 351 patients was selected using systematic random sampling. Structured questionnaires were used for data collection. Binary and multivariable logistic regression models were used to analyze the association between intention to use SMS reminders to promote drug adherence among individuals with diabetes and related factors.

Results

A total of 333 respondents, with a 94.87% response rate, were interviewed for this study. The majority of respondents, 66.4 % (95% CI [61.9–71.2]), expressed an intention to use SMS-based reminder services to promote their drug adherence. Age < 45 years (AOR = 5.73, 95% CI [2.07–15.73]), higher educational level (AOR = 3.03, 95% CI [1.16–7.90]), type of diabetes (AOR = 3.71, 95% CI [1.16–7.90]), oral medication users (AOR = 2.99, 95% CI [1.42–6.32]), SMS as a preferred medium for communication (AOR = 2.86, 95% CI [1.17–7.00]) were deemed to be important variables linked to intention to use SMS reminders to promote drug adherence among individuals with diabetes.

Conclusion

The findings suggest the majority of individuals with diabetes have intention to use SMS reminders to enhance adherence. This result indicates the potential for utilization of SMS reminders to enhance adherence to diabetic medications. Furthermore, the findings highlight the importance of tailored interventions that take into account patient characteristics and preferences as factors that influence intention when designing such an intervention.

Introduction

Medication adherence refers to the extent to which a patient’s behavior aligns with the prescribed dosing regimen, including the timing and intervals of intake, and is essential for the effectiveness of prescribed treatments (Dehdari & Dehdari, 2019; Gast & Mathes, 2019). Suboptimal adherence to anti-diabetic medications contributes to 30% to 50% of treatment failures and complications, leading to recommendations for an adherence rate of 80% or higher for individuals with diabetes (Torres-Robles et al., 2018; Kirkman et al., 2015). A systematic review conducted in Ethiopia indicated that the total pooled prevalence of adherence to anti-diabetic medication among adult individuals with diabetes mellitus(DM) was below the recommended levels at 69.5% (Yazew, Walle & Azagew, 2019). This and similar studies conducted in sub-Saharan Africa have shown that there is suboptimal adherence to diabetes medication, and thus, this has posed a major challenge in the management of patients (de Murwanashyaka et al., 2022; Ali, Alemu & Sada, 2017).

The mobile sector is communications industry’s fastest-growing segment in low-income nations and has a wide geographic coverage (Nigussie et al., 2021). According to Ethiopian Demographics Health Survey (EDHS) conducted in 2016, the overall population using mobile phones was reported as 56% with 88% and 47% in urban and rural households respectively (Central Statistical, Agency (CSA), 2016). Among the different mobile phone features, short message service (SMS) is one of the most popular forms of mobile communication and is the most widely used mobile data transfer system on a global scale (Mekonnen et al., 2021).

Mobile health (mhealth) has the potential to circumnavigate the practical limitations of physical presence in medical offices and is also widely available opening the possibility for frequent use in scheduling alerts and reminders (Huo et al., 2019). Of the different platforms of mhealth, for low and middle income counties, text messaging is a cost-effective measure for promoting health and has been found to positively impact increase in adherence to medications (Owolabi & Goon, 2019; Pandey et al., 2017).

Studies have indicated improved adherence among individuals who receive text reminders for medications. One comprehensive meta-analysis found that the probability of adhering to medication schedules approximately doubles when individuals receive reminders via text messages on their mobile phones (Thakkar et al., 2016). A systematic review done to assess the effectiveness of SMS messages to improve medication adherence among a wide range of chronic conditions, including diabetes, found that 18 out of 29 studies using text messaging interventions (TMI) reported statistically significant improvements in medication adherence rates or biomarkers (Park, Howie-Esquivel & Dracup, 2014). Furthermore, this study indicated that most reviewed studies reported high participant satisfaction with receiving SMS messages for use in promoting adherence. Overall, various studies have indicated that the use of tailored or personalized messages has been found to positively affect medication adherence (Park, Howie-Esquivel & Dracup, 2014; Garofalo et al., 2016; Basu et al., 2021).

Moreover, studies that were conducted on the receptiveness of individuals living with diabetes towards embracing mhealth services in the sub-Saharan Africa context have produced comparable results. A study conducted on willingness to use mhealth interventions for diabetes self-care in Gondar, Ethiopia, has shown that 70.5% of respondents were willing to use mhealth services (Jemere et al., 2019); these finding aligns with comparable studies carried out at Mizan Tepi University, Ethiopia and three tertiary healthcare institutions in Southwest Nigeria, where 59.1% and 72.6% of individuals diagnosed with diabetes mellitus, respectively, demonstrated a readiness to adopt mhealth-based services (Jemere et al., 2019; Olamoyegun et al., 2020).

The rapidly increasing number of mobile phone users in Addis Ababa, coupled with the suboptimal drug adherence among individuals with diabetes and positive user feedback to similar mhealth interventions, suggests that SMS-based mobile health initiatives hold strong potential for enhancing drug adherence and self-care in this population (Ali, Alemu & Sada, 2017; Coleman et al., 2020; Thomsen et al., 2019). While recognizing the critical significance of mhealth and its role in diabetes care and management, there remains a paucity of evidence regarding the intention of individuals with diabetes to utilize mhealth interventions, such as SMS reminder services to enhance their medication adherence in Addis Ababa, Ethiopia. This study aims to assess the intention to utilize mhealth interventions, specifically SMS reminder services, among individuals with diabetes in Addis Ababa, Ethiopia, to improve medication adherence, and to examine the associated factors influencing their intention.

Materials and Methods

Study area, design, and period

This is an institutional cross-sectional study conducted at Zewditu Memorial Hospital, which is one of the largest and oldest specialized hospitals under the administration of Addis Ababa Health Bureau (AAHB) located in Addis Ababa, Ethiopia. Services for individuals with diabetes are provided by various departments in the hospital. The diabetic referral and regular medical out-patient departments (OPDs) are part of the hospital’s diabetic follow-up clinics. The clinics offer services Monday through Friday with follow-up appointments as frequently as every one or 2 weeks and a maximum appointment every 3 months. Diabetes-related care services are provided at the hospital by general practitioners, specialist internists, and nursing staff. Data collection was conducted from February 06, 2023, to March 27, 2023.

Study population

The study population comprises adult individuals with diabetes on follow-up at the hospital for 3 or more months in the diabetic referral and regular medical OPD with access to a mobile phone. This study included individuals over 18 years old with either type 1 or type 2 diabetes who were receiving follow-up care at the hospital. Individuals diagnosed with gestational diabetes mellitus, individuals with diabetes that are seriously ill and require emergency admission, individuals with mental illnesses that significantly impeded their ability to communicate effectively, and those individuals who refused to consent were excluded from this study.

Sample size determination, sampling techniques, and procedures

The sample size was determined using both single and double population proportion formulas. The largest sample size was obtained using the single population proportion formula, n = (Z3/2)2 * p(q)/d2, where n represents the required sample size, Z is the standard normal distribution value corresponding to α/2, which is 1.96, p is the proportion of individuals with diabetes intending to use mhealth interventions for self-care at 70.5%, based on a study from the University of Gondar (Jemere et al., 2019), q is the proportion not intending to use mhealth, and d is the margin of error, 0.05. Including a 10% non-response rate estimated based on a previous study in Ethiopia (Jemere et al., 2019), the final calculated sample size was 351. Although the sample size was also calculated using the double proportion formula with assumptions of a 95% confidence level, 80% power, and a one-to-one ratio between case and control proportions, the single proportion formula yielded the largest sample size and was therefore selected as the final sample size. The study participants were selected using a systematic random sampling technique and proceeded in order of their appointments until an adequate sample size was reached (Fig. 1).

Figure 1 Proportional allocation of the study participants in the diabetic referral and regular medical out-patient departments at Zewditu Memorial Hospital.

Study variables

The outcome variable of interest is Intention to use SMS reminders to improve drug adherence among diabetic patients. The variables, Perceived Ease of Use (PEOU), Perceived Usefulness (PU), and Attitude towards Usage (AU) are derived from the Technology Acceptance Model (TAM), and indicate the essential components that influence a user’s acceptance of a system (Shroff, Deneen & Ng, 2011). Additionally, socio-demographic characteristics, environmental factors, clinical and behavioral factors and cell phone use pattern and privacy were all considered as predictor variables for the outcome of interest in this study and were adopted from other similar studies (Jemere et al., 2019; Bogale et al., 2022; Shropshire, Allen & Johnston, 2007).

Operational definitions

Intention to use

Intention to use refers to the extent to which a person has consciously formulated plans to perform or avoid performing a specific behavior in the future (Brezavšček, Šparl & Žnidaršič, 2016). A study comparing validated measures of intention has indicated that using a single-item measure of intention can sufficiently predict the implementation of the intended action it aims to assess (Fishman, Lushin & Mandell, 2020). Thus, the intention to use SMS reminders for improvement of drug adherence was assessed by a single item on a five-point Likert score, measured ranging from strongly disagree (1), disagree (2), undecided (3), agree (4), strongly agree (5). The Kolmogorov–Smirnov (K-S) test was used to check the normality of data. The K-S test and comparison of the histogram of the data to a normal probability curve was made. The K-S test revealed a p-value less than 0.05, suggesting that the data was not normally distributed. Thus, the median value was considered as a cut-off point to classify the intention. Values of intention that are equal to or greater than the median value of ‘4’ in the 5-point Likert scale were considered as a “yes,” while values less than the median value were considered a "no".

Perceived usefulness (PU)

Perceived usefulness (PU) refers to the user’s perception that utilizing this mobile health system will enhance their drug adherence (Abboodi, Abid & Mahmood, 2018). The perceived usefulness of SMS reminders for improvement of drug adherence was measured by five items on a five-point Likert scale from strongly disagree to agree. The K-S test and comparison of the histogram of the data to a normal probability curve was made. The K-S test revealed a p-value less than 0.05, indicating that the data was not normally distributed. Thus, the median value was considered as a cut-off point to classify the perceived usefulness. Values of PU equal to or greater than the median value of ‘4’ in the 5-point Likert scale were considered as a “yes,” while values less than the median value were considered a “no”.

Perceived ease of use (PEOU)

Perceived ease of use (PEOU) refers to the extent to which a person believes that using this mobile health system would require minimal effort (Lai, 2017). Perceived ease of use of SMS reminders for improvement of drug adherence measured by six items on a five-point Likert scale. The K-S test revealed a p-value less than 0.05, indicating that the data was not normally distributed thus the median value was considered as a cut of point to classify the PEOU. Values of PEOU that are equal to or greater than the median value of ‘4’ in the 5-point Likert scale were considered as a “yes”, while values less than the median value were considered a “no”.

Attitude towards usage (AU)

Attitude towards usage (AU) refers to the extent of the evaluative impact a person associates with using the target system (Marikyan & Savvas, 2023). The K-S test revealed a p-value less than 0.05, indicating that the data was not normally distributed; thus, the median value was considered a cut-off point to classify the AU. Values of AU that are equal to or greater than the median value of ‘4’ in the 5-point Likert scale are considered as a “yes,” while values less than the median value are considered a “no”.

Drug adherence

Drug adherence refers to whether or not patients take their medications as directed (Ho, Bryson & Rumsfeld, 2009). This is measured by a single-item question stating, ‘Over the past 7 days, how many times did you miss a dose of any of your diabetic medication?’ (Wu et al., 2014). Those who missed a dose more than or equal to once a week were considered non-adherent; those with no missed doses were considered adherent.

Data collection procedures and personnel

Data was collected through face-to-face interviews using structured questionnaires adopted from prior studies (Jemere et al., 2019; Shroff, Deneen & Ng, 2011; Bogale et al., 2022; Shropshire, Allen & Johnston, 2007). Data was collected on participants’ sociodemographic factors, clinical and behavioral factors, cell phone use pattern, perceived usefulness, perceived ease of use, attitude towards usage, and intention to use SMS reminders to promote diabetes medication adherence.

Data quality control and assurance

To evaluate the understandability and the applicability of the questionnaire, a pretest was conducted on 10% of the calculated sample size 1 week before the main study. Following the analysis of the pretest, ambiguous or unclear questions were re-assessed, after which specific questions and wordings were modified. Four data collectors, each with a BSc degree in nursing and prior experience in quantitative data collection, were engaged in the study. A full day of training was conducted the day before data collection began, covering the study objectives, methods, ethical principles and field procedures. The principal investigator also carried out on-site supervision, and feedback was provided to data collectors promptly. Data collectors used alcohol-based hand sanitizers and facemasks during data collection to prevent COVID-19, as it was still a public health concern during the conduction of the data collection.

Data analysis procedures

Data from the questionnaire was coded and entered using the Epi info v7 database, and then data was exported to Excel, and from there it was exported to IBM SPSS V26.0 for analysis. Data was summarized using frequencies and percentages for categorical variables and mean with standard deviation for the numerical variables as required. Binary logistic regression was chosen because the outcome variable was dichotomous (i.e., had two categories) and this analysis was conducted to assess the relationships between independent and dependent variables. The bivariate analysis was conducted at a 0.25 level of significance to screen for potentially significant independent variables. Before conducting the multivariable logistic regression, model multicollinearity was checked using a variable inflation factor (VIF) with a cutoff point set to <10. Then, multivariable logistic regression was run by including these variables from the bivariate analysis. Hosmer and Lemeshow goodness of fit was computed, and the model was adequate with a p-value of 0.25. To measure the presence and strength of association between the independent variables and intention to use SMS reminders to promote drug adherence, odds ratio, p-value, and 95% CI for the regression coefficient (β) were used. From these variables, those with p-value < 0.05 were considered to be significantly associated with the intention to use SMS-based reminder services to promote drug adherence.

Ethics approval

Appropriate ethical clearance and a supportive letter were obtained from Addis Continental Institute of Public Health with Ref.no: Addis Continental Institute of Public Health (ACIPH)-MPH-055/15 on 14/12/2022 and Addis Ababa Health Bureau with Ref.no: 6343/227 on 22/12/2022. Ethical guidelines of the Declaration of Helsinki for studies involving human participants were followed during the conduct of the study. Participants gave informed written consent to participate in the study before taking part. Anonymity of study subjects was maintained to ensure confidentiality of information obtained. Consent forms were stored in a restricted and confidential environment to safeguard participant privacy.

Results

Sociodemographic and environmental characteristics of the respondents

Three hundred thirty-three individuals with diabetes who had a personal mobile phone participated in the study with a response rate of 94.87%. More than half of the participants were female, 190 (57.1%), with a mean age of 52.17 (SD = 14.2) years. The majority of participants were married 236 (70.9%) and live together in one house with other individuals, 301 (90.4%). Regarding educational status, 144 (43.2%) participants had a primary or secondary level of education, with 136 (40.8%) of the respondents, employed (Table 1).

Table 1 Socio-demographic and environmental characteristics of individuals with diabetes at Zewditu memorial hospital Addis Ababa, Ethiopia, 2023 (N = 333).

Variable	Frequency (n)	Percentage	
Sex	Male	143	42.9	
Female	190	57.1	
Age	≤45	106	31.8	
46–60	131	39.3	
≥61	96	28.8	
Marital status	Single/separated/widowed/divorced	97	29.1	
Married	236	70.9	
Educational status	No formal education	79	23.7	
Primary or secondary education	144	43.2	
Higher education	110	33.0	
Employment/occupation	Unemployed	130	39.0	
Employed	136	40.8	
Retired	67	20.1	
Living arrangement	I live alone	32	9.6	
I live with others	301	90.4	

Clinical and behavioral attributes

The majority of respondents have type 2 diabetes 254 (76.3%), with more than half of the respondents having one or more co-morbidities, 207 (62.2%). With regard to the route of medication intake, 150 (45%) of the participants reported taking medication via the oral route only. The majority of respondents, 237 (71.2%), reported that they did not miss any medication dose over the past 7 days, indicating good adherence. More than half of the respondents, 211 (63.4%), did not actively use reminder services to take medication. However, among those who used reminders, 76 (62.3%) reported using their mobile phone as a reminder (Table 2).

Table 2 Clinical and behavioral attributes of individuals with diabetes at Zewditu Memorial Hospital Addis Ababa, Ethiopia, 2023 (N = 333).

Variable	Frequency(n)	Percentage	
Type of diabetes	Type I	79	23.7	
Type II	254	76.3	
Time since diagnosis	<12 months	19	5.7	
>12 months	314	94.3	
Co-morbidities	No	126	37.8	
Yes	207	62.2	
Hypertension (n = 207)	No	44	21.3	
Yes	163	78.7	
Cardiovascular diseases (n = 207)	No	189	91.3	
Yes	18	8.7	
Dyslipidemia (n = 207)	No	173	83.6	
Yes	34	16.4	
Route of medication intake	Injection	133	39.9	
Pill	150	45.0	
Both	50	15.0	
Missed medication dose over the past 7 days (Adherence)	Poor adherence (once or more)	96	28.8	
Good adherence (never)	237	71.2	
Use of reminder mechanisms	No	211	63.4	
Yes	122	36.6	
Type of reminder mechanism	Pillbox/written schedule/watch alarm	46	37.7	
Mobile phone reminders	76	62.3	
The habit of taking addictive substances	No	312	93.7	
Yes	21	6.3	
Blood glucose monitoring frequency	>Once a month	189	56.8	
≤Once a month	144	43.2	

Mobile usage patterns

With regard to mobile usage patterns among the respondents, the majority, 173 (52%) reported owning a smartphone, and more than two-thirds had their cell phone with them at all times, 281 (84.4%). Verbal communication was the preferred way of communication among the respondents of this study, 234 (70.3%). The majority of participants could read or send text messages via their mobile devices, 226 (67.9%). More than two-thirds of the respondents, 251 (75.4%), were willing to pay for text messaging services for medication reminders, and from those willing to pay, almost all, 245 (97.6%), were only willing to pay less than 30 birr or 0.54 USD per month.

Technology acceptance component factors

Of the total respondents of the study, 230 (69.1%) reported that they believe using SMS reminders as a reminder for medication intake will be easy for them to use and adopt easily indicating a higher level of perceived ease of use. Additionally, 227 (68.2%) participants reported that they believe SMS for medication reminders will be useful, suggesting perceived usefulness. Furthermore, a majority of participants, 232 (69.7%), reported a favorable attitude towards using SMS for medication reminders, indicating a positive attitude towards usage.

Intention to use SMS-based reminders

Among the respondents of this study, 221 (66.4%) of the participants (95% CI [61.9–71.2]) expressed the intention to utilize SMS-based reminder services to enhance their drug adherence if the service was made available.

Association between different factors and Intention to use SMS reminders to promote drug adherence

Results of the bivariate analysis indicated that sex, age, marital status, educational status, employment status, accommodation, type of diabetes, co-morbidities, previous use of any reminder mechanisms, type of mobile phone, frequency of mobile phone availability, preferred way of communication, willingness to pay for SMS services, perceived ease of use(PEU), perceived usefulness (PU), attitude towards usage (AU) were all found to be significantly associated with intention to use SMS-based mhealth interventions to promote drug adherence at a p-value threshold of 0.25. Although route of mediation intake was not a statistically significant predictor of outcome in the bivariate analysis, because it is a variable of scientific importance, it was included into the multivariable logistic regression model. However, on the multivariable logistic regression model, after adjusting for other covariates, age, education, type of diabetes, route of medication intake, and preferred way of communication, were found to have a statistically significant association with intention to use SMS reminder service to promote drug adherence among diabetic patients.

Accordingly, after adjusting for the other covariates, the odds of having the intention to use SMS reminders to promote drug adherence was 5.73 times higher among those aged less than 45 as compared to those above 61 years of age (AOR = 5.73, 95% CI [2.09–15.73], p = 0.001). Respondents from the age group 46-60 had 2.74 times higher odds of intention to use SMS reminders to promote drug adherence as compared to the older population (AOR = 2.74, 95% CI [1.27–5.91] p = 0.01). Regarding education, the odds of intention to use SMS reminders to promote drug adherence was 3.03 times higher in those that have higher education as compared with those that do not have any form of formal education (AOR = 3.03, 95% CI [1.16–7.90], p = 0.02). Similarly, respondents that had a primary or secondary education level of education had a 2.25 times higher odds of intention to use these SMS-based reminder services as compared to those that do not have any form of formal education (AOR = 2.25, 95% CI [1.08–4.69], p = 0.03). With regard to the type of diabetes, the odds of having the intention to use SMS reminders to promote drug adherence was 3.71 times higher among those with type I diabetes as compared to those with type II diabetes (AOR = 3.71, 95% CI [1.16–7.90], p = 0.02).

Similarly, with regards to the route of medication, the odds of having the intention to use SMS reminders to promote drug adherence was 2.99 times higher among those who use oral routes of medication administration as compared to those who use injection as a route of medication (AOR = 2.99, 95% CI [1.42– 6.32], p = 0.001). In addition to this, those who preferred text messaging as a route of communication have 2.86 times more odds of having the intention to use SMS-based reminder services for promoting drug adherence as compared to those who preferred a verbal way of communication (AOR = 2.86, 95% CI [1.17–7.00] p = 0.02) (Table 3).

Table 3 Bivariate and multivariable logistic regression analysis of factors associated with intention to use short messaging service (SMS) among individuals with diabetes at Zewditu Memorial Hospital Addis Ababa, Ethiopia, 2023 (N = 333).

Variable	Intention to use SMS for medication reminders	OR (95%CI)	AOR (95%CI)	p-value	
No	Yes	
Sex	Male	39	104	1	1		
Female	73	117	0.60 [0.38–0.96]	1.36 [0.62–2.97]	0.44	
Age	<45	15	91	6.88 [3.49–13.54]	5.73 [2.09–15.73]*	0.00*	
46–60	46	85	2.09 [1.22–3.59]	2.74 [1.27–5.91]	0.01*	
>61	51	45	1	1		
Marital status	Single/separated/widowed/divorced	14	43	1	1		
Married	74	162	0.71 [0.37–1.38]	1.44 [0.70–2.97]	0.32	
Education	No formal education	52	27	1	1		
Primary or secondary education	42	102	4.68 [2.59–8.42]	2.25 [1.08–4.69]	0.03*	
Higher education	18	92	9.84 [4.96–19.56]	3.03 [1.16–7.90]	0.02*	
Employment	Unemployed	56	74	1	1		
Employed	31	105	2.56 [1.51–4.36]	1.39 [0.63–3.10]	0.42	
Retired	25	42	1.27 [0.69–2.33]	2.17 [0.81–5.79]	0.12	
Accommodation	I live alone	15	17	1	1		
I live with others	97	204	1.86 [0.89–3.87]	1.77 [0.61–5.14]	0.29	
Type of DM	Type I	11	68	4.08 [2.06–8.09]	3.71 [1.38–9.94]	0.01*	
Type II	101	153	1	1		
Comorbidities	No	28	98	1	1		
Yes	84	123	0.42 [0.25–0.69]	0.88 [0.45–1.73]	0.87	
Route of medication intake	Injection	44	89	1	1		
Pill	49	101	1.02 [0.62–1.68]	2.99 [1.42–6.32]	0.00*	
Both	19	31	0.81 [0.41–1.59]	2.52 [1.02–6.25]	0.05	
Previous use of any reminder mechanisms	No	77	134	1	1		
Yes	35	87	1.43 [0.88–2.31]	1.26 [0.66–2.37]	0.48	
Type of phone	Non-smartphone	81	79	1	1		
Smartphone	31	142	4.69 [2.86–7.72]	1.99 [0.95–4.17]	0.07	
Frequency of having a mobile phone	Not always	33	19	1	1		
Always	79	202	4.44 [2.39–8.27]	1.42 [0.64–3.15]	0.39	
Preferred way of communication	Verbal	104	130	1	1		
Text	8	91	9.10 [4.22–19.60]	2.86 [1.17–7.00]	0.02*	
Notes:

* Indicates statistical significance (p < 0.05).

DM, diabetes mellites.

Discussion

The results indicated that 66.4 % (95% CI [61.9–71.2]) of the respondents expressed an intention to use SMS-based reminder services to help improve their drug adherence. Age, education, type of diabetes, route of medication intake, and preferred way of communication were significantly associated with intention to use SMS-based reminders in the promotion of drug adherence.

The findings of this study were found to be consistent with similar studies conducted at Mizan Tepi University Teaching Hospital, Ethiopia 59.1% (Bogale et al., 2022), Gondar university hospital, Ethiopia 70.5% (Jemere et al., 2019), and three teaching hospitals in south west Nigeria, 72.6% (Olamoyegun et al., 2020). Similar findings were also found among epileptic patients at Gondar University Hospital 68.8% (Senay et al., 2019). The findings of this study were lower than those reported in a study conducted in Bangladesh, where the willingness to use mhealth interventions was 99.6% (Ismal et al., 2016). Additionally, our results indicated a slightly lower proportion of respondent willingness compared to two similar mhealth interventions conducted in Ethiopia, which reported proportions of 71.4% and 74.5%, respectively (Walle et al., 2023; Firdisa et al., 2023). However, the proportion observed in this study is higher than in studies conducted among patients on antiretroviral treatment in North West Ethiopia, where, 50.9%, were willing to receive text message medication reminders (Kebede et al., 2015). Furthermore, the findings of this study are higher than those of studies conducted in 2016 and 2017, which reported that 56.7% of respondents in the United States and 50% in Tokyo, Japan, respectively were willing to receive text message reminders among individuals with diabetes (Shibuta et al., 2017; Humble et al., 2016). The variation in participants’ socioeconomic backgrounds, levels of familiarity with digital technologies, the perception of current self-care practices as being sufficient, privacy and the feeling of burden associated with using SMS reminder services could be implicated in the differences (Bogale et al., 2022; Kebede et al., 2015; Shibuta et al., 2017).

The current study shows that younger individuals with diabetes mellitus are more likely have intention to use SMS services for medication reminders to improve their drug adherence as compared to older diabetic individuals. This can be attributed to the higher likelihood that younger adults are more tech-savvy, adapt to using technology, and use a wide variety of technologies compared to older ones (Olson et al., 2011). This finding aligns with previous studies conducted in northwest and central Ethiopia (Kebede et al., 2015; Endebu et al., 2019).

According to the findings of this study, being educated is also positively associated with intention to use SMS reminders to promote drug adherence. Education is one factor that fosters a more positive outlook on new technology and improves one’s capacity to learn in new situations and adapt (Blut & Wang, 2020). The findings of this study are in line with two similar studies conducted in northwestern Ethiopia (Jemere et al., 2019; Kebede et al., 2015).

This study also found that being a type I diabetic was positively associated with the intention to use SMS reminders for diabetic drug adherence. This can be explained in relation to age, in which the significant majority of individuals with type I DM are in the younger age group as compared to type II DM patients; this becoming a substantial predictor as the younger age group has more intention as compared to the older ones.

In addition, those taking pills for their medications were more open to using SMS-based reminder services than people who take insulin via injection or both routes of medication administration. This aligns with a study conducted in northwest Ethiopia in which those taking pills had more receptiveness towards using mobile health interventions (Jemere et al., 2019). In low middle income countries (LMIC), individuals with type 2 diabetes are often started on insulin later in the disease progression due to clinical inertia and patient resistance (Basu & Sharma, 2018). As individuals remain in the disease state longer, they often develop additional comorbidities, such as hypertension and high cholesterol, which increase their medication burden (Cicek et al., 2021). This would increase the need for such a reminder service. However, those who take insulin may have developed a stronger routine and awareness around their treatment, given the structured nature of insulin administration, thus requiring less reminder support compared to individuals taking oral medications.

Furthermore, the study also found that individuals who prefer text messaging as a mode of communication have a better intention to use SMS reminders for medication adherence. This could be because people who prefer text messaging are more accustomed to utilizing SMS and would be more willing to incorporate this technology into their everyday lives, making it a convenient choice for their medication reminders. The finding of this study is consistent with a study conducted at Vanderbilt University Medical Center, United States, in which the majority of individuals with elevated blood pressure chose text messaging as a delivery route for reminders and medical information (Nelson et al., 2022). The findings of another study conducted in Pakistan also indicate that more than half of the respondents implied that they would be open to using text messaging to deliver medical services (Iftikhar et al., 2019).

Limitations

This study adopted a cross-sectional design, which limits its ability to establish temporal relationships among variables. The potential for recall bias exists, as participants relied on their memory to provide information about past events. Additionally, social desirability bias may have influenced participants’ responses. This study was conducted at a specific hospital in a particular city in Ethiopia; the findings may not be fully generalizable to other settings. Therefore, Caution should be exercised when inferring these results to different regions or populations.

Conclusion

The findings suggest that a majority of respondents intend to use SMS reminders as a tool to promote adherence. Factors such as age, education, type of diabetes, route of medication intake, and preferred communication method were identified as notable influences associated to use SMS reminders for promoting adherence among individuals with diabetes. Furthermore, these results highlight the importance of tailored interventions considering patient characteristics and preferences.

Supplemental Information

Supplemental Information 1 Dataset.

Supplemental Information 2 Codebook for the dataset.

The authors would like to thank the Addis Continental Institute of Public Health (ACIPH) for technical support. The authors would also like to thank all study participants, data collectors, and supervisors involved in this study.

Abbreviations

AAHB Addis Ababa Health Bureau

AU Attitude Towards Usage

DM Diabetes Mellitus

EDHS Ethiopia Demographic Health Survey

HIV Human Immune Virus

LMIC Low- and Middle-Income Countries

PEOU Perceived Ease of Use

OPD Outpatient Department

PU Perceived Usefulness

RMOPD Regular Medical Outpatient Department

SMS Short Messaging Service

TAM Technology Acceptance Model

TMI Text Messaging Interventions

WHO World Health Organization

Additional Information and Declarations

Competing Interests

Author Contributions

Human Ethics

Data Availability

The authors declare that they have no competing interests

Solomon Alem conceived and designed the experiments, performed the experiments, analyzed the data, prepared figures and/or tables, authored or reviewed drafts of the article, and approved the final draft.

Hanna Gulema analyzed the data, authored or reviewed drafts of the article, and approved the final draft.

The following information was supplied relating to ethical approvals (i.e., approving body and any reference numbers):

Addis Continental Institute of Public Health (ACIPH) - Ref No: ACIPH-MPH-055/15.

Addis Ababa Health Bureau- To conduct the study at Zewditu Memorial Hospital Ref.no: 6343/227.

The following information was supplied regarding data availability:

The raw dataset is available in the Supplemental File.

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
