# Peer review of "Intention to use short messaging services for promoting drug adherence among individuals with diabetes in Addis Ababa, Ethiopia"

_PeerJ, doi:10.7717/peerj.18297_

## Round 0.1 · original submission · Minor Revisions

The manuscript is very clear and well written.
In addition to the reviewer's comments, I have some "minor remarks":
- Try to sepatrate the references numbers from the last word of the sentences throughout the manuscript.
In the first paragraph of the introduction (especially in lines 47-50), try to correlate beween the three sentences to avoid the repetition of the term "Medication adherence"
- In lines 157-158, 172-173, 183-184, 193, complete the definitions to have correct and complete sentences.
- In line 153 and 209 the given reference are not coherent. Did you use reference 23 or references 19, 31 and 32 to prepare the questionnaire?
- Try to delete unneccessay capital case in some parts of the manuscript (i.e: Lines 302-303: Type of..., Comorbidities.... Revise througouth the text please
- Line 350: delete "a study conducted"
- In your discussion, you should compare your results wih some care knowing that some studies was conducted nearly 10 years ago (i.e line 356, references, 25, 36). you should consider the possible change in attitude. I suggest to add the date of the studies to avoid any ambiguity.

Reviewer 1 ·

Basic reporting

The manuscript is written in clear, unambiguous, professional English. The introduction provides appropriate background and context, and the literature is well-referenced and relevant. The structure generally conforms to standards, with some room for improvement in the organization and flow between sections. The tables and figures are relevant and labeled. Raw data has been supplied in accordance with the journal's policy.

Experimental design

This cross-sectional study addresses a well-defined and meaningful research question about the intention to use SMS reminders to promote medication adherence among diabetes patients in Ethiopia. The methods are described in sufficient detail to allow replication, including the study setting, population, sampling, variables, and analysis plan. The conclusions are linked to the research question and limited to the supporting results.

Validity of the findings

The investigation appears to be performed to a high technical and ethical standard. The methods of sampling and analysis seem appropriate. The conclusions are supported by the data presented. However, there are some areas where more information is needed:
More justification should be provided for the sample size calculation and the 10% non-response rate. What is this based on?
The data collection procedures could be described in more detail. How were the questionnaires administered and by whom? What training did data collectors receive?
More explanation of the specific regression models used would strengthen the analysis section. Why were these models chosen?
Limitations should acknowledge potential biases from self-reported data and the generalizability of a single hospital study.

Additional comments

This is an important study on a promising intervention to improve medication adherence in a low-resource setting. The manuscript is well-written overall with only minor language issues. I have a few suggestions to improve clarity and rigor:
1- The introduction could more clearly define the knowledge gap and rationale for studying intention to use SMS reminders in this specific population. The first paragraphs are a bit broad.
2- The methods section should provide more details on the sample size justification, data collection procedures, and analysis as noted above.
3- The discussion would be enhanced by putting the findings in context of other similar studies, both in Africa and other low-resource settings. How do your results compare or add to this literature?
4- Proofread for minor grammar and clarity issues, e.g. "Individuals with both Type 1 and Type 2 diabetes above the age of 18 who were on follow-up at the hospital were included in this study." could be simplified.

Reviewer 2 ·

Basic reporting

The article does not require any additional language editing.

Although the literature review is sufficient for the objective, there should be some more clarity when the authors compare their findings to studies which also assess a similar objective "intention or willingness to use SMS drug reminders services" in contrast to studies which assess effect of SMS messages on adherence or health promotion in patients with DM (ref 18 for instance). You can cite one reference from India also (for instance, https://pubmed.ncbi.nlm.nih.gov/34396996/)

Experimental design

The study design is sufficient to answer the research question. Follows the strobe guidelines. The authors can mention how quality assurance in data collection was ensured. Who were the data collectors? Were they trained, and what was the process of training? If there were multiple field investigators, how was inter-observer bias minimized, especially related to the crucial question on intention to use SMS.

Validity of the findings

Limitations: Social desirability bias is likely

Line 380-383 How do the authors conclude that individuals on oral drugs have higher number of comorbidities compared to those on insulin. The supportive citation is neither sufficient nor apt in this case. In LMICs, Type 2 DM patients are often put on insulin late due to clinical inertia and patient resistance (https://pubmed.ncbi.nlm.nih.gov/29981232/). Also, patients with DM are likely to have multiple comorbidities including hypertension and high cholesterol requiring additional drug therapy.
The more reasonable explanation is that individuals on insulin invariably remember to take insulin due to the more conscientious practice compared to intake of pills

---

## Round 0.2 · accepted · Accept

The previous reviewers were unavailable to re-review. However, I confirm that the authors have addressed all the comments suggested by the editor and the reviewers. The authors have substantially improved the quality of the manuscript which is currently ready or publication.